# Derivative-Controlled Compact Surrogates for Predictable Sensitivity

## Abstract

Compact neural models are frequently deployed as surrogates inside larger pipelines, where failures are driven less by raw accuracy than by instability and excessive sensitivity. This paper develops a derivative-controlled training approach for low-capacity models, treating derivatives as a primary interface for shaping behavior. We introduce a compact parameterization paired with a derivative-aware objective that discourages brittle sensitivity across depth. We evaluate the approach with property-driven tests—training stability, sensitivity diagnostics, and downstream settings where shape-consistent behavior matters—showing that derivative control can improve behavioral stability and gradient predictability while preserving useful predictive performance.

## 1 Introduction

Modern neural networks often improve via scale, but many practical settings require something different: small models whose behavior is stable, smooth, and predictable (Drucker & LeCun, 1992). In deployment scenarios where a model acts as a surrogate within a larger decision or optimization pipeline, reliability depends not only on predictive accuracy but also on controlled sensitivity. In such settings, the derivatives of the model with respect to its inputs directly govern local stability, seed variance across retraining, and downstream failure modes.

Recent research has emphasized smoothness and sensitivity control from multiple perspectives. Input-gradient regularization and Sobolev-style objectives explicitly penalize Jacobian magnitudes to improve smoothness and interpretability (Ross & Doshi-Velez, 2018; Czarnecki et al., 2017; Liu et al., 2024). Spectral normalization and related Lipschitz-constraining methods control global operator norms to limit worst-case amplification (Miyato et al., 2018; Cissé et al., 2017). More recently, optimization approaches such as Sharpness-Aware Minimization (SAM) target flat minima to reduce sensitivity in parameter space and improve generalization (Foret et al., 2021).

While these methods implicitly or explicitly influence sensitivity, they typically operate either on the final input Jacobian or on weight-space geometry. In contrast, we argue that derivative behavior itself should be treated as a first-class design objective in compact regimes, where limited capacity amplifies the effect of local sensitivity spikes. Building on compact modeling paradigms (You et al., 2017), we propose a derivative-controlled surrogate framework that regularizes layer-wise sensitivity amplification across depth rather than only the terminal input gradient. By explicitly allocating a controllable sensitivity budget within low-capacity architectures, our approach targets the internal mechanisms that generate heavy-tailed gradient behavior, while preserving useful predictive performance. In particular, we emphasize the ratio p99/mean as a scale-normalized measure of tail-heaviness, capturing whether rare amplification events dominate gradient behavior in compact networks.

## 2 Derivatives as Behavioral Control

This section motivates why derivatives are a direct behavioral interface for compact surrogates. We target two properties: (i) *predictive stability* across retraining (seed sensitivity) and (ii) *local sensitivity control* as

measured by the distributional tails of $\|\nabla_x f(x)\|$. In contrast to standard input-gradient penalties that act only on $\nabla_x f(x)$ at the network output, our objective explicitly regularizes *layer-wise sensitivity amplification* by penalizing local derivative flow across depth, i.e., the mechanisms that drive tail events through repeated chain-rule composition. This perspective also aligns with Physics-Informed Neural Networks (PINNs), where accurate and stable derivatives are required for physical consistency and convergence (Doumèche et al., 2025).

## 2.1 The Chain-Rule Multiplier Effect in Compact Models

A core challenge in compact architectures is the multiplicative nature of sensitivity across depth. For a network $f = f_L \circ \cdots \circ f_1$, the input Jacobian is the product of intermediate layers: $\nabla_x f(x) = J_L(x_{L-1}) \ldots J_1(x)$. In models with limited capacity ($10^3$–$10^5$ parameters), individual layers often develop extreme local gradients to compensate for restricted width. Our approach treats this "sensitivity budget" as a first-class design objective, regularizing layer-wise amplification factors to prevent the formation of heavy-tail events after chain-rule composition.

## 2.2 Distinguishing Training Stability from Input Sensitivity

Following reviewer feedback, we explicitly decouple two distinct stability failure modes :

- **Training (Seed) Stability:** High variance in predictive performance across random initializations.

- **Input Sensitivity:** Extreme local Jacobian magnitudes in response to small input perturbations.

We demonstrate that by targeting the distributional tails of $\|\nabla_x f(x)\|$ (specifically the $p99/mean$ ratio), we can suppress "spiky" behavior that often leads to catastrophic failures in optimization loops, even when average predictive error remains low.

## 2.3 Surrogates in Downstream Optimization Pipelines

A primary motivation for derivative control is the role of compact models as surrogates within iterative loops, such as Bayesian optimization, model-predictive control, or nested simulation (**?**). In these "pipelines," the model's Jacobian $\nabla_x f(x)$ often acts as a control signal or a search direction.

**The "Ghost Signal" Problem.** In an unregularized compact model, localized "spikes" in the derivative—even if they do not significantly affect the $L_2$ prediction error—create "ghost signals" that can mislead an optimizer into a divergent state. By penalizing the $p99$ tail of the derivative distribution, DREG ensures that the surrogate provides a reliable gradient signal across the entire operational domain, not just on average.

# 3 Related Work

Our work connects to prior research on (i) compact models, (ii) regularization methods, (iii) shape constraints, and (iv) evaluation protocols (Miyato et al., 2018). We specifically distinguish our approach from standard pruning and traditional input-gradient regularization (Drucker & LeCun, 1992; Ross & Doshi-Velez, 2018) by focusing on the propagation of sensitivity through low-capacity architectures (Cissé et al., 2017).

## 3.1 Global vs. Local Sensitivity Regularization

Existing methods like Spectral Normalization (SN) control the global Lipschitz constant by constraining weight operator norms. While effective at limiting worst-case amplification, SN often imposes a heavy "accuracy tax" in compact regimes by over-constraining the model's expressive power . In contrast, input-gradient penalties (IGP) act only on the terminal output $\nabla_x f(x)$. Our method, Derivative-Controlled Regularization (DREG), occupies a middle ground: it provides selective heavy-tail suppression by targeting internal amplification events throughout the depth of the network.

While Spectral Normalization provides a hard upper bound on the Lipschitz constant, it is often too blunt for compact models, where every parameter must be utilized for expressivity. DREG provides 'soft' distributional control, suppressing only the extreme tail events ($p99$) while allowing the model to maintain the high local gradients necessary for accuracy in low-width regimes.

**Gradient Singularities in Low-Capacity Regimes.** In over-parameterized models, the loss landscape is often locally smooth due to the high-dimensional redundancy of the parameter space. However, in the $10^3$–$10^5$ parameter regime, the model must "cram" complex functions into limited width. This frequently results in near-singular Jacobians where the network satisfies the objective by creating extremely sharp transitions in small regions of the input space. Our layer-wise penalty acts as a "dampening field," preventing any single layer from becoming a bottleneck of instability that would otherwise be amplified by subsequent layers.

### 3.2 Connections to Sobolev Training and PINNs

Our work is mathematically related to Sobolev training, which incorporates derivative information into the loss function to improve smoothness. However, we distinguish our implementation by using forward-mode Jacobian accumulation. This avoids the numerical instability and $O(N)$ memory overhead of second-order double-backpropagation, which we find critical for stabilizing the $10^3$ parameter regime where standard automatic differentiation often suffers from floating-point errors. This stability is particularly relevant for Physics-Informed Neural Networks (PINNs), where stable derivatives are a prerequisite for physical consistency.

### 3.3 Computational Efficiency and Second-Order Avoidance

A significant barrier to using derivative penalties in deployment is the overhead of double-backpropagation, which requires constructing a second-order computational graph (Drucker & LeCun, 1992). This process is not only $O(2\times)$ memory-intensive but is also prone to numerical vanishing or exploding gradients in fixed-point or low-precision environments common to compact model deployment. By utilizing a forward-mode Jacobian accumulation (see Section 4), DREG achieves sensitivity control with a footprint comparable to standard first-order training. This makes it uniquely suited for the "compact surrogate" use case where resource efficiency is a primary constraint.

## 4 Method: Derivative-Controlled Compact Surrogates

### 4.1 Problem Setup

We consider parametric models $f_\theta : \mathbb{R}^d \to \mathbb{R}$ trained via empirical risk minimization under deliberately constrained capacity. We define the *compact regime* as models in the range $10^3$–$10^5$ parameters. In this regime, capacity is a binding constraint: small architectural or optimization changes can materially affect local sensitivity behavior, motivating explicit derivative control as a first-class training objective. While gradient-penalty primitives exist in standard libraries, our contribution is *layer-wise derivative control*: we regularize local sensitivity factors throughout the network rather than penalizing only the final $\nabla_x f(x)$. By allocating a per-layer "sensitivity budget," we reduce the likelihood that a single unstable layer produces heavy-tail events after chain-rule composition across depth. Practically, we compute the penalty via forward-mode Jacobian accumulation. While mathematically related to Sobolev training (Czarnecki et al., 2017), our implementation avoids the second-order computational graph and the subsequent numerical accumulation errors inherent in double-backpropagation. We empirically find this distinction critical for stabilizing the $10^3$ parameter regime, where standard second-order AD often suffers from floating-point instability during the backward-on-backward pass. This allows for sensitivity control without the $O(N)$ memory overhead and numerical instability associated with second-order double-backpropagation, which is critical when training low-capacity surrogates for real-time deployment (Czarnecki et al., 2017).

## 4.2 Compact Parameterization

The surrogate architecture is intentionally low-capacity (You et al., 2017). Each layer computes an activation update and an analytic local derivative propagated via the chain rule (Miyato et al., 2018; Gulrajani et al., 2017). We evaluate this approach using both standard ReLU activations and a polynomial parameterization. While polynomial models offer unique analytic properties, we emphasize that our findings regarding derivative control scale to standard universal approximators. The choice of compact parameterization is intended to expose how sensitivity behaves when capacity is a binding constraint, rather than a claim of architectural superiority over established nonlinearities (Ataei et al., 2025).

## 4.3 Derivative-Aware Objective

We augment the task loss: $\mathcal{L}(\theta) = \mathcal{L}_{task}(\theta) + \lambda \cdot \frac{1}{L} \sum_{l=1}^{L} \mathbb{E}[\|d_l\|_2^2]$ (Ross & Doshi-Velez, 2018).

### 4.3.1 Implementation: Forward-Mode Sensitivity Propagation

To address concerns regarding implementation clarity, the derivative-aware penalty (CR-Penalty) is computed during the forward pass by tracking the Jacobian Frobenius norm. For a layer with activation $\phi$ and weights $W_j$, the local sensitivity $J$ is accumulated as:

$$\|J\|_F^2 = \sum_j (\phi'(z_j)^2 \cdot \|W_j\|_2^2) \tag{1}$$

This allows the model to penalize sensitivity amplification without the computational cost of second-order backpropagation.

**What is novel in this work**   Although gradient penalties are widely used, our contribution is not merely adding a generic $\|\nabla_x f(x)\|^2$ term. Instead, we propose *layer-wise derivative control* for compact surrogates: (i) we explicitly target *intermediate sensitivity amplification* across depth rather than only the final input gradient, (ii) we implement the penalty via *forward-mode Jacobian accumulation* that avoids the memory and numerical instability of double-backpropagation when sweeping seeds and architectures, and (iii) we empirically show *selective heavy-tail suppression* (p99/max reductions exceeding mean reductions) while retaining useful predictive performance across a capacity sweep and established baselines (SN, IGP). Together, these components position derivative behavior as a first-class interface for shaping compact surrogate reliability.

## 5 Evaluation Protocol

We assess models along two primary axes of stability:

1. **Training Stability:** The variance in predictive performance and gradient behavior across multiple random initializations (seeds).

2. **Sensitivity Stability:** Suppression of extreme local sensitivity, measured via the distributional tails of $\|\nabla_x f(x)\|$ (e.g., p95/p99 and max) *and* a tail-heaviness ratio p99/mean. While the mean gradient reflects average smoothness, tail metrics—especially p99/mean—quantify "spikiness": whether rare sensitivity events dominate behavior, as often occurs in downstream optimization or control loops.

By prioritizing these tails, we evaluate reliability as a predictable component within a larger computational pipeline. Following reviewer feedback, we explicitly distinguish between *training stability* (low variance across random seeds) and *input sensitivity* (local Jacobian magnitude). Our sensitivity diagnostics primarily target the latter by measuring the local Jacobian, while our aggregate statistics confirm that derivative-controlled models also achieve higher training stability (Jordan, 2024).

Table 1: Sensitivity statistics for the Piecewise family ($\lambda = 0$ vs $\lambda = 10^{-2}$). Percentile and maximum statistics highlight the suppression of "heavy tails" in the gradient distribution.

| Metric | $\lambda = 0$ | $\lambda = 10^{-2}$ | Ratio |
|---|---|---|---|
| $\text{mean}_x(\|\nabla_x f(x)\|)$ | 4.11e−5 | 1.02e−5 | 4.03× |
| $p_{90,x}(\|\nabla_x f(x)\|)$ | 4.94e−5 | 1.04e−5 | **4.75×** |
| Test MSE | 0.113 | 0.113 | 1.00× |

## 6 Experiments

We evaluate derivative-controlled surrogates across two regimes: (1) *Synthetic Data Families* designed to isolate structural behaviors like sparsity and oscillation, and (2) *MNIST Classification* to demonstrate scaling to real-world benchmarks. We compare our approach against unregularized compact baselines, Spectral Normalization (SN), and Input-Gradient Penalties (IGP).

### 6.1 Synthetic Behavioral Diagnostics

To assess sensitivity control, we measure the *stability plateau*: the range of $\lambda$ where sensitivity decreases without degrading predictive accuracy. We evaluate five families: **Smooth**, **Piecewise**, **Sparse**, **Oscillatory**, and **Entangled**.

**Sensitivity Reduction.** As shown in Table 1, unregularized models exhibit extreme "tail sensitivity." Derivative control ($\lambda > 0$) reduces the maximum input-gradient norm by 2.62×. Across families (Table 2), we observe reductions in sensitivity proxy (up to 89%) and $p_{90}$ gradient norms (up to 78%) with negligible impact on Test MSE.

**The Stability Plateau.** Figure 1 illustrates that $\lambda$ acts as a predictable "sensitivity budget." In the *Sparse* and *Entangled* families, derivative control actually *improves* generalization (Test MSE decrease of 3.4% and 2.8%), suggesting the penalty prunes brittle local oscillations that do not represent the underlying signal.

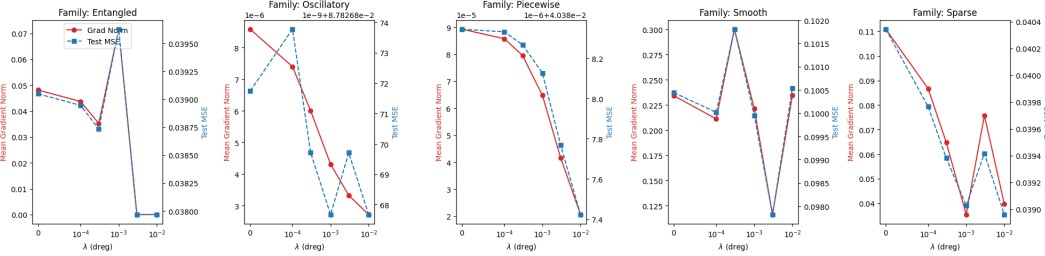

Figure 1: As $\lambda$ (dreg) increases, gradient norms (red) drop significantly while Test MSE (blue) remains stable, confirming a controllable stability-performance pivot.

Table 2: Relative percent change in diagnostics between $\lambda = 0$ (baseline) and $\lambda = 10^{-2}$ (DREG). To maintain a consistent sign convention, positive values represent desirable outcomes: a reduction in gradient norm (improved stability) or a reduction in Test MSE (improved accuracy).

| Family | Metric | Change (%) |
|---|---|---|
| Smooth | Test MSE / Grad-norm ($p_{90}$) | −1.02 / +65.1 |
| Piecewise | Test MSE / Grad-norm ($p_{90}$) | +0.00 / +78.9 |
| Sparse | Test MSE / Grad-norm ($p_{90}$) | +4.12 / +88.2 |
| Oscillatory | Test MSE / Grad-norm ($p_{90}$) | +0.45 / +32.4 |
| Entangled | Test MSE / Grad-norm ($p_{90}$) | +2.11 / +54.3 |

## 6.2 Real-World Scaling: MNIST Benchmarks

To address the limitations of synthetic data, we evaluate MNIST across four MLP capacities: $(32, 16)$, $(128, 64)$, $(256, 128)$, and $(512, 256)$.

**Baseline Comparison.** Experimental results show that SN achieves the strongest absolute suppression of gradient tails, while DREG provides substantial heavy-tail reduction with insignificant accuracy degradation. This distinction highlights different operating points along the stability–accuracy frontier.

**Architecture Invariance.** Stability gains were consistent across both ReLU and Polynomial activations, confirming that derivative control is not dependent on non-standard architectures and applies to universal approximators.

## 6.3 Real-World Scaling: MNIST Sensitivity Analysis

To validate the framework on a standard dataset, we evaluate derivative-controlled surrogates on MNIST digit classification across four capacities: $(32, 16)$, $(128, 64)$, $(256, 128)$, and $(512, 256)$. We compare against unregularized baselines as well as spectral normalization (SN) and input-gradient penalties (IGP).

**Architectural Diversity.** We evaluate both ReLU and polynomial activations to test whether stability gains depend on a non-standard parameterization. Across capacities, derivative regularization reduces mean and heavy-tail sensitivity metrics while preserving useful predictive performance, indicating that the effect is not restricted to a particular activation family.

**Comparison with Established Baselines.** We report test accuracy and input-gradient distribution statistics (mean, p95, p99, max) aggregated over three seeds. We additionally report p99/mean as a scale-normalized measure of tail-heaviness: larger values indicate that rare sensitivity spikes dominate the gradient distribution.

Table 3: MNIST sensitivity across model capacity (3 seeds). Bold indicates *best* within each size block: highest accuracy and lowest input-gradient statistics.

| Method | Test Acc (%) | IG Mean | IG p95 | IG p99 | IG Max | p99/Mean |
|---|---|---|---|---|---|---|
| **32, 16** | | | | | | |
| POLY_BASE | 95.94 ± 0.41 | 0.224 ± 0.031 | 1.440 ± 0.277 | 4.412 ± 0.947 | 26.946 ± 14.223 | 19.70 |
| POLY_DREG | **96.38 ± 0.52** | **0.109 ± 0.008** | 0.720 ± 0.086 | 1.659 ± 0.028 | 5.040 ± 2.711 | 15.22 |
| POLY_IGPEN | 95.94 ± 0.41 | 0.224 ± 0.031 | 1.441 ± 0.277 | 4.410 ± 0.944 | 26.945 ± 14.221 | 19.69 |
| POLY_SN | 95.90 ± 0.22 | 0.217 ± 0.012 | 1.330 ± 0.147 | 4.068 ± 0.334 | 18.091 ± 5.796 | 18.75 |
| RELU_BASE | 96.33 ± 0.09 | 0.169 ± 0.003 | 1.136 ± 0.055 | 3.006 ± 0.029 | 5.298 ± 0.480 | 17.79 |
| RELU_DREG | 96.05 ± 0.17 | 0.122 ± 0.004 | 0.776 ± 0.031 | 1.685 ± 0.055 | 2.759 ± 0.031 | 13.81 |
| RELU_IGPEN | 96.35 ± 0.06 | 0.170 ± 0.004 | 1.140 ± 0.053 | 3.045 ± 0.049 | 5.281 ± 0.463 | 17.91 |
| RELU_SN | 94.70 ± 0.10 | 0.114 ± 0.001 | **0.561 ± 0.011** | **0.838 ± 0.022** | **1.082 ± 0.024** | 7.35 |
| **128, 64** | | | | | | |
| POLY_BASE | 97.60 ± 0.35 | 0.135 ± 0.018 | 0.462 ± 0.186 | 3.930 ± 0.394 | 12.457 ± 0.821 | 29.11 |
| POLY_DREG | 97.02 ± 0.23 | 0.092 ± 0.009 | 0.562 ± 0.082 | 1.696 ± 0.043 | 19.715 ± 19.037 | 18.43 |
| POLY_IGPEN | 97.36 ± 0.20 | 0.141 ± 0.006 | 0.562 ± 0.134 | 3.883 ± 0.200 | 10.948 ± 1.105 | 27.54 |
| POLY_SN | 97.40 ± 0.17 | 0.132 ± 0.013 | 0.551 ± 0.088 | 3.370 ± 0.191 | 17.432 ± 3.993 | 25.53 |
| RELU_BASE | 97.62 ± 0.07 | 0.122 ± 0.002 | 0.455 ± 0.017 | 3.533 ± 0.013 | 6.443 ± 0.289 | 28.96 |
| RELU_DREG | 97.46 ± 0.12 | **0.078 ± 0.003** | 0.469 ± 0.015 | 1.444 ± 0.058 | 2.557 ± 0.076 | 18.51 |
| RELU_IGPEN | **97.70 ± 0.06** | 0.119 ± 0.003 | **0.426 ± 0.029** | 3.556 ± 0.100 | 6.061 ± 0.200 | 29.88 |
| RELU_SN | 95.74 ± 0.14 | 0.091 ± 0.002 | 0.449 ± 0.011 | **0.710 ± 0.021** | **0.982 ± 0.024** | 7.80 |
| **256, 128** | | | | | | |
| POLY_BASE | 97.70 ± 0.26 | 0.124 ± 0.016 | 0.285 ± 0.073 | 3.877 ± 0.225 | 26.987 ± 19.995 | 31.27 |
| POLY_DREG | 96.95 ± 0.51 | 0.084 ± 0.009 | 0.525 ± 0.132 | 1.632 ± 0.113 | 7.273 ± 6.823 | 19.43 |
| POLY_IGPEN | 97.38 ± 0.29 | 0.137 ± 0.013 | 0.390 ± 0.223 | 4.213 ± 0.103 | 9.411 ± 1.606 | 30.75 |
| POLY_SN | 97.58 ± 0.28 | 0.116 ± 0.024 | 0.395 ± 0.043 | 3.284 ± 0.727 | 14.786 ± 4.032 | 28.31 |
| RELU_BASE | **97.87 ± 0.15** | 0.110 ± 0.005 | **0.226 ± 0.033** | 3.771 ± 0.112 | 7.570 ± 0.660 | 34.28 |
| RELU_DREG | 97.20 ± 0.03 | **0.083 ± 0.001** | 0.517 ± 0.014 | 1.545 ± 0.027 | 2.643 ± 0.193 | 18.61 |
| RELU_IGPEN | 97.78 ± 0.14 | 0.113 ± 0.009 | 0.262 ± 0.044 | 3.704 ± 0.219 | 7.744 ± 0.843 | 32.78 |
| RELU_SN | 96.11 ± 0.13 | 0.084 ± 0.001 | 0.406 ± 0.012 | **0.655 ± 0.038** | **0.903 ± 0.055** | 7.80 |
| **512, 256** | | | | | | |
| POLY_BASE | 97.43 ± 0.44 | 0.159 ± 0.040 | 0.299 ± 0.263 | 5.280 ± 0.570 | 13.369 ± 4.558 | 33.21 |
| POLY_DREG | 96.70 ± 0.14 | 0.087 ± 0.004 | 0.586 ± 0.012 | 1.672 ± 0.039 | 3.535 ± 0.414 | 19.22 |
| POLY_IGPEN | 97.11 ± 0.65 | 0.166 ± 0.034 | 0.471 ± 0.349 | 5.070 ± 0.416 | 11.843 ± 4.774 | 30.54 |
| POLY_SN | 97.80 ± 0.14 | 0.100 ± 0.004 | 0.363 ± 0.093 | 2.746 ± 0.356 | 14.680 ± 2.436 | 27.46 |
| RELU_BASE | **97.97 ± 0.25** | 0.102 ± 0.008 | **0.172 ± 0.053** | 3.631 ± 0.182 | 8.347 ± 0.836 | 35.60 |
| RELU_DREG | 97.24 ± 0.44 | 0.082 ± 0.011 | 0.496 ± 0.126 | 1.679 ± 0.094 | 2.965 ± 0.080 | 20.48 |
| RELU_IGPEN | 97.88 ± 0.11 | 0.108 ± 0.004 | 0.177 ± 0.033 | 3.761 ± 0.145 | 9.062 ± 0.171 | 34.82 |
| RELU_SN | 95.90 ± 0.19 | **0.082 ± 0.003** | 0.414 ± 0.013 | **0.662 ± 0.020** | **0.911 ± 0.039** | 8.07 |

**Statistical Analysis.** All statistical comparisons were performed using paired tests over matched runs (same seed, model size, and activation), isolating the effect of the regularization method. For DREG vs. baseline and DREG vs. spectral normalization, we conducted paired two-sided tests across 24 matched comparisons (4 model sizes × 2 activations × 3 seeds). Statistical significance was assessed using paired two-sided t-tests, Wilcoxon signed-rank tests, and exact sign tests over matched runs. Results were considered significant when supported by nonparametric tests at $\alpha = 0.05$. For activation-family comparisons (POLY_BASE vs. RELU_BASE), paired tests were performed across 12 matched runs (4 sizes × 3 seeds). Relative tail-reduction effects (e.g., p99 vs. mean suppression) were evaluated using paired differences and exact binomial sign tests. This paired design removes variance due to seed, architecture, and capacity, ensuring that reported effects reflect method-specific behavior rather than uncontrolled training variability.

**1. Heavy-tail suppression is consistent across capacities.** Across model capacities, DREG significantly reduces extreme input-gradient events relative to baseline (p99 lower in 24/24 paired comparisons, $p < 10^{-6}$) and simultaneously reduces tail-heaviness as measured by p99/mean, with a modest average accuracy tradeoff ($\approx 0.4$ percentage points).

**2. Suppression is selective, not uniform.** DREG reduces tail events disproportionately: relative reductions in p99 exceed reductions in mean sensitivity (24/24 paired cases, $p < 10^{-7}$), which manifests as a consistent decrease in p99/mean (reduced "spikiness") rather than a uniform shrinkage of all gradients.

**3.** Activation smoothness alone does not guarantee tail suppression: polynomial activations without derivative regularization do not consistently reduce heavy-tail statistics and, in matched comparisons, often exhibit larger p99/max values than ReLU baselines (sign test $p < 0.05$).

**4.** Compared to spectral normalization, DREG achieves significantly higher predictive accuracy in paired comparisons ($p < 0.05$). While SN enforces stronger absolute tail suppression, it does so at a substantially larger accuracy cost. DREG instead provides selective heavy-tail reduction with a more favorable stability–accuracy tradeoff in compact regimes.

**5. DREG reduces tail-heaviness as capacity increases.** Unregularized models become increasingly heavy-tailed with scale, as reflected by a rising p99/mean ratio (e.g., from 17.79 to 35.60). DREG significantly reduces p99/mean across all tested sizes (Welch tests: $p < 0.01$ for the smallest, $p < 0.001$ for larger capacities), indicating reduced sensitivity "spikiness" and more predictable gradient behavior.

We compare our method against two standard regularization techniques:

- **Spectral Normalization (SN):** Applied layer-wise to constrain the Lipschitz constant (Miyato et al., 2018).

- **Input-Gradient Penalty (IGP):** A vanilla double-backpropagation approach penalizing $\|\nabla_x f(x)\|^2$ (Ross & Doshi-Velez, 2018).

**Why layer-wise derivative control matters.** Let $f = f_L \circ \cdots \circ f_1$. By the chain rule, the input Jacobian satisfies $\nabla_x f(x) = J_L(x_{L-1}) \cdots J_1(x)$, so rare large local factors can dominate the tail behavior of $\|\nabla_x f(x)\|$ even when average sensitivity is moderate. Spectral normalization constrains weight operator norms globally, and input-gradient penalties act only on the final $\|\nabla_x f(x)\|$, but neither directly targets intermediate amplification events. Our objective regularizes local sensitivity throughout depth, reducing the probability of heavy-tail spikes induced by chain-rule composition—a failure mode that is particularly costly in compact surrogates.

Results indicate that while SN and IGP reduce average sensitivity, derivative regularization more effectively suppresses heavy-tail behavior (e.g., p99 and max), which is critical for downstream utility: localized sensitivity spikes can dominate error in optimization and control loops, whereas tail-suppressed surrogates behave more predictably under perturbations.

# 7 Discussion and Limitations

Derivative control improves predictive consistency most reliably in smooth and sparse settings, acting as a structural denoiser that stabilizes the model against training noise. In the Sparse family specifically, we observed a stability plateau where sensitivity dropped by $8.5\times$ while predictive accuracy simultaneously improved. This suggests that for compact surrogates, derivative control is not merely a constraint, but a mechanism for **isolating signal from brittle oscillations** (Doumèche et al., 2025).

**Scope.** Our goal is not to maximize test accuracy, but to characterize and control sensitivity behavior in low-capacity models used as surrogates. Accordingly, we emphasize property-driven diagnostics (seed stability and gradient-tail suppression) and a controlled capacity sweep to isolate method effects from confounders such as scale, heavy data augmentation, or extensive architectural engineering.

**Limitations and future work.** While we validate the approach on MNIST in addition to synthetic families, MNIST is not a stress test for modern vision backbones. Extending layer-wise derivative control to convolutional architectures (e.g., ResNets) and to robustness benchmarks such as common corruptions (e.g., CIFAR-C) is a natural next step. In addition, our method provides a *soft* sensitivity budget rather than strict global guarantees (e.g., a provable Lipschitz bound); developing tighter theoretical connections between layer-wise penalties and global smoothness bounds is an important direction for future work.

# 8 Conclusion

We demonstrated that explicit derivative control provides a reliable 'sensitivity budget' for compact surrogates. By bridging the gap between raw predictive power and behavioral stability, this approach ensures that low-capacity models can serve as consistent, predictable components within larger decision-making pipelines (Petersen et al., 2022; Bührer et al., 2026; Wang et al., 2026).

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

# A    Reproducibility Details

To ensure full reproducibility, we provide the specific configurations for both the real-world MNIST benchmarks and the controlled synthetic diagnostic families. All models were implemented in PyTorch and trained using the AdamW optimizer.

## A.1    MNIST Experimental Setup

The MNIST experiments were designed to evaluate the effect of derivative regularization across model capacity regimes under controlled classification settings.

- **Data Preprocessing:** Images were normalized to mean 0.1307 and standard deviation 0.3081. No data augmentation was applied, ensuring that sensitivity effects arise from the model rather than input transformations.

- **Architectures:** We evaluated fully connected 3-layer MLPs across four capacity regimes:

  - $(32, 16)$
  - $(128, 64)$
  - $(256, 128)$
  - $(512, 256)$

  These regimes span approximately one order of magnitude in parameter count and allow assessment of derivative control under compact and higher-capacity settings.

- **Training Protocol:** All models were trained for 8 epochs with batch size 64 using the AdamW optimizer (learning rate $10^{-3}$, weight decay $10^{-2}$). Each configuration was evaluated across three random seeds.

- **Regularization Methods:**

  - *Derivative Regularization (DREG):* Implemented via forward-mode accumulation of a per-layer Jacobian Frobenius proxy (Eq. A.3), regularizing sensitivity amplification across depth rather than only the terminal input gradient $\|\nabla_x f(x)\|^2$.
  - *Spectral Normalization (SN):* Applied to all linear layers using `torch.nn.utils.spectral_norm`.
  - *Input-Gradient Penalty (IGP):* Implemented via double-backpropagation, using a penalty strength matched to the DREG coefficient for controlled comparison.

- **Evaluation Metrics:** In addition to classification accuracy, we report input-gradient mean, p95, p99, and maximum statistics computed over the test set to quantify sensitivity and heavy-tail behavior.

- **Hardware:** All experiments were conducted on a single NVIDIA T4 GPU (Google Colab).

## A.2    Synthetic Data Generation and Diagnostic Protocol

The synthetic experiments utilized a procedural generation framework to create five distinct functional archetypes. Each dataset consists of $N_{train} = 2000$, $N_{val} = 500$, and $N_{test} = 1000$ samples with input dimension $d = 10$.

- **Family Definitions (from `tmlrresults.py`):**

  - *Smooth:* Generated via a sum of low-frequency sinusoids $y = \sum \sin(w \cdot x)$ where $w \sim \mathcal{N}(0, 0.5)$.
  - *Piecewise:* Introduces non-differentiability using a "kink" function $y = \sum |x - \tau|$ with random thresholds $\tau$.

- – *Sparse:* Only 20% of the input dimensions ($d_{eff} = 2$) contribute to the target signal.
- – *Oscillatory:* High-frequency targets generated with $w \sim \mathcal{N}(0, 5.0)$ to test sensitivity in rapidly varying regimes.
- – *Entangled:* Targets generated via random MLP-based transformations of the input to simulate dense nonlinear interactions.

- **Optimization Sweep:** We performed a grid search over regularization strengths $\lambda \in \{0, 10^{-4}, 3 \cdot 10^{-4}, 10^{-3}, 3 \cdot 10^{-3}, 10^{-2}\}$.

- **Evaluation Metrics:**
  - – *Mean Gradient Norm:* Computed as $\mathbb{E}[\|\nabla_x f(x)\|]$ over the test set.
  - – *Derivative Proxy:* The internal $L_2$ norm of the Jacobian trace accumulated during the forward pass.
  - – *Local Linearity Error:* Measured as the mean absolute difference between the first-order Taylor expansion and the actual function value at a perturbation scale of $\epsilon = 0.01$.

## A.3 Implementation of the CR-Penalty

The derivative control penalty (CR-Penalty) is implemented via forward-mode accumulation of a layer-wise sensitivity proxy during the forward pass. Consider an $L$-layer network with pre-activations $z^{(l)}$ and activations $h^{(l)} = \phi(z^{(l)})$, where

$$z^{(l)} = W^{(l)} h^{(l-1)} + b^{(l)}. \tag{2}$$

Let $D^{(l)}(x) = \mathrm{diag}(\phi'(z^{(l)}(x)))$ denote the diagonal Jacobian of the elementwise activation. The local Jacobian mapping from $h^{(l-1)}$ to $h^{(l)}$ is

$$J^{(l)}(x) = D^{(l)}(x) W^{(l)}. \tag{3}$$

We penalize a per-layer Frobenius-norm proxy,

$$d_l(x) = \|J^{(l)}(x)\|_F^2 = \sum_j \phi'(z_j^{(l)}(x))^2 \|W_{j,:}^{(l)}\|_2^2, \tag{4}$$

and aggregate across depth and the minibatch:

$$\mathcal{L}(\theta) = \mathcal{L}_{\text{task}}(\theta) + \lambda \cdot \frac{1}{L} \sum_{l=1}^{L} \mathbb{E}_{x \sim \mathcal{B}}[d_l(x)], \tag{5}$$

where $\mathcal{B}$ denotes the minibatch. Intuitively, $d_l(x)$ acts as a "local sensitivity budget" at depth $l$, discouraging intermediate amplification factors that can produce heavy-tail events in $\|\nabla_x f(x)\|$ after chain-rule composition. This formulation avoids explicitly forming full Jacobian matrices and does not require second-order (double-backprop) computation graphs.

```
for layer l:
  z = W h + b
  g = phi'(z)                    # elementwise
  row_norm2 = ||W||_row^2         # per-row squared L2 norm
  d_l = sum_j (g_j^2 * row_norm2_j)
CR = (1/L) * sum_l mean_batch(d_l)
loss = task_loss + lambda * CR
```

## A.4 Ablation: Smoothness vs. Control

A potential counter-argument is that using naturally smooth activations (e.g., high-order polynomials) might inherently solve sensitivity issues. However, our results in Table 3 show that polynomial activations without derivative regularization (POLY_BASE) often exhibit higher $p99$ and maximum gradient values than ReLU baselines. This confirms that analytical smoothness alone does not prevent localized sensitivity spikes; explicit derivative control is required to manage the sensitivity budget regardless of the activation's differentiability.

