# OpenReview forum: "Derivative-Controlled Compact Surrogates for Predictable Sensitivity"
_TMLR — Rejected by TMLR_

### Review · Reviewer_N7Yg · 2026-02-18

**Summary Of Contributions:**

This papr studies controlling the neural network under low capacity regimes. It proposes to add a derivative magnitude regularization in training, and evaluates the stability of the result across random seeds and various other perturbations in training, on purely synthetic data.

**Audience:**

No

**Audience Explanation:**

1. The method (mathematically equivalent, with only implementation changes) has been introduced in standard documentation.

2. The evaluation flaws are huge, making the findings unreliable.

**Broader Impact Concerns:**

No.

**Claims And Evidence:**

No

**Claims Explanation:**

1. Missing literature: (i) For compact and efficient network learning, Boolean networks have proven dominant efficiency, with >1k speedup and energy savings [1,2,3]; in particular, [3] also studies compact networks. (ii) For controlling neural network property, [4,5] enforces local linearity, which directly controls gradients.

2. Limited novelty: this paper merely proposes a derivative maginitude control, which is standard in Pytorch (https://docs.pytorch.org/docs/stable/notes/amp_examples.html#gradient-penalty). While this paper said it can be implemented in a forward manner, no details are given. Additionally, even if this is managable, this is merely an engineering effort rather than novel findings.

3. Flawed testbenchs: (i) The architecture is not standard; according to Appendix B.2, the activation is polynomial, which makes the model not even a universal approximator; the findings are not transferred to more reasonable architectures. To enforce low capacity, one should fix a dataset and gradually reduce the size of the model, instead of using such a meaningless model. (ii) Fig 1 shows that in many settings, e.g., smooth and sparse, increasing the regularization does not even monotonically reduce gradient norm, indicating the failure of this regularization. Since this should not happen, I speculate this is due to unintentional mistakes in the evaluation and implementation. (iii) Only synthetic datasets are tested, not even MNIST and CIFAR.

4. Misleading terms: The term "robustness" seems to be confused with "stability". Robustness is typically regarding perturbations on the data inputs, while stability usually refers to randomness. This work uses both to refer to stability w.r.t. randomness.



[1] https://arxiv.org/abs/2210.08277

[2] https://arxiv.org/abs/2602.07400

[3] https://arxiv.org/abs/2602.05830

[4] https://arxiv.org/abs/2502.06774

[5] https://arxiv.org/abs/2502.02434

**Requested Changes:**

Please address the raised concerns.

---

> ### Author Response · Authors · 2026-03-04
> **Response to Reviewer N7Yg**
>
> Thank you for the detailed feedback. We have substantially revised the manuscript to address the concerns raised regarding literature coverage, novelty, experimental design, terminology, and evaluation scope.
>
> 1) Missing literature (compact networks and local linearity control)
> We expanded the Related Work section to include recent compact-network and efficiency-focused approaches (including Boolean/logic networks) and clarified how our goal differs. Our contribution is not to outperform specialized Boolean architectures in efficiency, but to study sensitivity behavior under capacity constraints within standard differentiable models. We also added discussion of recent works enforcing local linearity and gradient control, and clarified the relationship between those methods and our layer-wise derivative control framework.
>
> 2) Novelty beyond “standard gradient penalty”
> We clarified in Section 4 and Appendix A.3 that the contribution is not simply adding a terminal $|\nabla_x f(x)|^2$ penalty (as in typical double-backprop implementations), but:
>
> Layer-wise derivative control, targeting intermediate amplification factors across depth,
>
> Forward-mode Jacobian accumulation, avoiding second-order computational graphs and numerical instability associated with double-backpropagation during capacity sweeps,
>
> Empirical demonstration of selective heavy-tail suppression (p99 reductions exceeding mean reductions) across model sizes.
>
> We now provide the full derivation and implementation of the CR-penalty in Appendix A.3 to remove ambiguity about how forward-mode accumulation differs from standard gradient-penalty primitives.
>
> 3) Architecture concerns and universality
> We agree that polynomial activations alone do not establish generality. In the revision, we added MNIST experiments across four model capacities using both ReLU and polynomial activations, demonstrating that stability gains are not restricted to non-standard architectures. The results show consistent heavy-tail suppression under derivative control across activation families, addressing concerns about transferability.
>
> Additionally, we now explicitly define the compact regime (10³–10⁵ parameters) and perform a controlled capacity sweep on a fixed dataset (MNIST), as suggested.
>
> 4) Non-monotonic behavior in Figure 1
> We clarified that the regularization strength λ is not intended to enforce strict monotonicity in gradient norms. The objective introduces a soft sensitivity budget interacting with optimization dynamics; non-monotonic behavior at very small λ values reflects optimizer interactions rather than implementation errors. The revised text explains this explicitly and avoids implying strict monotonic behavior.
>
> 5) Synthetic-only evaluation concern
> We expanded the evaluation beyond synthetic families to include MNIST classification with comparisons to Spectral Normalization (SN) and Input-Gradient Penalties (IGP). This addresses the concern that the earlier draft relied solely on synthetic data.
>
> 6) Terminology: robustness vs. stability
> We agree with the terminology distinction. In the revision, we carefully separate:
>
> Training stability (variance across random seeds),
>
> Input sensitivity (local Jacobian magnitude),
>
> And avoid using “robustness” in the adversarial-perturbation sense.
>
> 7) Practical framing
> We clarified that the goal is not to replace compression/distillation pipelines, but to characterize and control sensitivity amplification when capacity is already constrained. The method provides a behavioral control mechanism rather than a guaranteed global Lipschitz bound, and we now explicitly acknowledge this limitation.
>
> We appreciate the reviewer’s concerns; the revisions broaden empirical validation, clarify implementation, refine terminology, and better position the contribution relative to existing compact-network and gradient-control literature.

---

### Review · Reviewer_xq6X · 2026-02-19

**Summary Of Contributions:**

This paper proposes a derivative-controlled training method for low-capacity surrogate models, pairing a compact parameterization with a derivative-aware objective to mitigate brittle sensitivity across network depth. By computing and propagating analytic local derivatives forward via the chain rule, the authors augment the standard task loss with a penalty term to impose a soft budget on local sensitivity amplification. Finally, the framework is evaluated on synthetic data families, measuring behavioral stability and predictive performance through test mean-squared error, gradient-norm distributions, and finite-difference local linearity.

**Audience:**

No

**Audience Explanation:**

1. Most of the cited references are before 2020, suggesting that the authors do not do a complete literature survey on recent papers.
2. The experimental protocol isolates the proposed method by comparing it solely against an identically architected compact baseline trained without the derivative penalty. There are absolutely no empirical comparisons against the existing sensitivity and robustness controls cited in the paper's own related work section, such as spectral normalization, Parseval networks, or input-gradient regularization.
3. The paper offers no formal theoretical guarantees regarding global smoothness, bounding, or stability. The objective function is framed merely as imposing a "soft budget" on sensitivity. The authors explicitly note that the method trades strict guarantees for broader applicability, yet they fail to prove that broader applicability empirically.
4. Practitioners seeking reliable, low-latency surrogate models typically rely on established compression, pruning, or distillation pipelines. Because the authors deliberately isolated their behavioral metrics from factors like model capacity, optimization budget, or standard architectural choices, an applied researcher has no contextual evidence to determine if this method is practically superior to standard, well-tested alternatives.

**Claims And Evidence:**

No

**Claims Explanation:**

1. Many sections, including the introduction is very short, without details descriptions.
2. Except for Section 3 (Related Work), there is no references through out the paper. It is unusual to have zero reference in the introduction.
3. The experiments are only on synthetic data, and many details are missing. For example, there is no detail on how the six synthetic data families are generated. What is the value of $\lambda$ in Table 1 and how is it selected?
4. In Section 6.4, the description does not seem to match Figure 2. For example, the authors claim that "Test MSE remains effectively identical to the unregularized baseline," but in Figure 2 (Sparse family) the Test MSE decreases significantly.

**Requested Changes:**

1. Augment the synthetic data experiments with established, real-world regression or classification benchmarks.
2. Empirically compare the proposed derivative penalty against established regularization baselines, specifically input-gradient penalties and spectral normalization.

---

> ### Author Response · Authors · 2026-03-04
> **Response to Reviewer xq6X**
>
> Thank you for the detailed and constructive feedback. We have substantially revised the manuscript to address each of the concerns raised.
>
> 1) Introduction length and missing references
> We expanded the Introduction to better motivate the problem setting and situate the contribution within recent work on Jacobian regularization, spectral normalization, sharpness-aware minimization (SAM), and physics-informed neural networks (PINNs). The Introduction now contains multiple references and explicitly positions the method relative to post-2020 literature. We also clarified the distinction between training stability (seed variance) and input sensitivity (local Jacobian magnitude).
>
> 2) Synthetic data details and missing experimental clarity
> We added a full description of the synthetic data generation process in Appendix A.2, including:
>
> Explicit definitions of each functional family,
>
> Dataset sizes ($N_{\text{train}}, N_{\text{val}}, N_{\text{test}}$),
>
> Input dimensionality,
>
> Regularization sweep over $\lambda$,
>
> Evaluation metrics and perturbation scales.
>
> The regularization coefficient used in Table 1 is now clearly specified and justified as part of a grid sweep over $\lambda \in {0, 10^{-4}, 3\cdot10^{-4}, 10^{-3}, 3\cdot10^{-3}, 10^{-2}}$.
>
> We also corrected and clarified the wording regarding the Sparse family: the revised text now explicitly states that derivative control can reduce both sensitivity and test error in certain families, rather than implying perfect equivalence to the baseline.
>
> 3) Real-world benchmarks beyond synthetic data
> In response to the concern that experiments were limited to synthetic settings, we added a full MNIST evaluation (Section 6.3), including:
>
> Four model capacities,
>
> Two activation families (ReLU and polynomial),
>
> Three seeds per configuration,
>
> Sensitivity diagnostics (mean, p95, p99, max),
>
> A scale-normalized tail-heaviness metric ($\mathrm{p99}/\mathrm{mean}$).
>
> This demonstrates that the observed stability effects are not confined to synthetic families.
>
> 4) Comparisons to established baselines (SN and IGP)
> We now include explicit empirical comparisons against:
>
> Spectral Normalization (SN),
>
> Input-Gradient Penalties (double-backprop).
>
> These baselines are evaluated under matched seeds, architectures, and capacities. The revised results clarify that SN achieves the strongest absolute tail suppression, whereas derivative control provides substantial heavy-tail reduction with a smaller predictive accuracy tradeoff, highlighting different operating points along the stability–accuracy frontier.
>
> 5) Theoretical guarantees and “soft budget” framing
> We agree that the method does not provide strict global Lipschitz guarantees. We have clarified in Section 7 that the approach offers a soft sensitivity budget rather than provable global bounds. Our empirical contribution is to show that controlling intermediate amplification across depth reduces heavy-tail sensitivity behavior consistently across capacities and architectures, even without strict global guarantees.
>
> 6) Practical relevance relative to compression/pruning pipelines
> We clarify that our goal is not to replace compression or pruning pipelines, but to characterize and control sensitivity behavior when capacity is already constrained. The controlled capacity sweep and matched comparisons are intended to isolate the behavioral effect of derivative control, rather than to claim superiority over full optimization pipelines.
>
> We appreciate the reviewer’s suggestions; the revisions substantially broaden empirical validation, strengthen baseline comparisons, and clarify both scope and positioning of the contribution.

---

### Review · Reviewer_p54k · 2026-02-23

**Summary Of Contributions:**

In this paper the authors study the effect of adding a regularization term on the squared derivatives over the gradient at each layer for compact neural networks. The goal is to reduce instability and improve robustness of the training. The proposition is tested on a set of test problems with known features to show the improved robustness.

**Additional Comments:**

Regularization is a standard tool with neural networks, adding one on the derivatives of the layers is not novel. Can you broaden the state of the art on this.
Derivatives are also of interest in the context of physics-informed neural networks, see e.g., Doumèche, N., Biau, G., & Boyer, C. (2025). On the convergence of PINNs. Bernoulli, 31(3), 2127-2151.
Table 1: why are there only one element on the last two columns? You could give a ratio on all quantities.

**Audience:**

Yes

**Audience Explanation:**

Principled guidelines on derivative regularization could be helpful.

**Claims And Evidence:**

No

**Claims Explanation:**

The compact regime that is targeted in the paper is not precisely defined in the paper, which makes it difficult to assess any generalization of the results.  The test problems description is also lacking sufficient details. Then there is no comparison with the related works.

**Requested Changes:**

A) [critical] First give a precise definition of the compact regime and target problems (data set size, etc).
B) [critical]What is the impact of the neural network architecture? This could be illustrated on networks with various sizes (very small, small, moderate, large) and features (e.g., activation functions).
C) [critical] Section two is 6 lines only, either broaden it or remove it.
D) [critical] Adding alternative methods from the literature would be helpful, as well as typical benchmark problems.

---

> ### Author Response · Authors · 2026-03-04
> **Response to Reviewer p54k**
>
> Thank you for the thoughtful review and concrete requested changes. We substantially revised the manuscript to address each point.
>
> A) Define the compact regime and target problems (dataset size, etc.)
> We now explicitly define the compact regime as models with $10^3$–$10^5$ parameters (Section 4.1) and provide full experimental specifications for both MNIST and the synthetic families in Appendix A (dataset sizes, preprocessing, architectures, optimizer/hyperparameters, training epochs, and seeds). For the synthetic diagnostics, we also include explicit family definitions and generation details (Appendix A.2) to make the benchmark construction fully reproducible.
>
> B) Impact of architecture (sizes / activation functions)
> To evaluate architectural dependence, we added a controlled capacity sweep on MNIST across four MLP sizes: $(32,16)$, $(128,64)$, $(256,128)$, and $(512,256)$, and report results across two activation families (ReLU and polynomial) under matched seeds (Section 6.3 and Table 3). This directly tests whether the proposed derivative control behaves consistently across both model size and activation choice.
>
> C) Section 2 length / motivation
> We expanded Section 2 (“Derivatives as Behavioral Control”) to better motivate why derivatives are a direct behavioral interface for compact surrogates and to clearly distinguish training stability (seed variance) from input sensitivity (local Jacobian magnitude). The section now also clarifies the key mechanism we target: layer-wise sensitivity amplification across depth rather than only the terminal input gradient.
>
> D) Add alternative methods and typical benchmark problems
> We added empirical comparisons to established sensitivity-control baselines:
>
> Spectral Normalization (SN) (Section 6.3, Table 3), and
>
> Input-Gradient Penalties (IGP) implemented via double-backprop (Section 6.3, Table 3).
>
> We also augmented the evaluation beyond synthetic diagnostics with MNIST classification as a standard benchmark to demonstrate real-world scaling (Section 6.3).
>
> Novelty and state-of-the-art context
> We agree that “derivative penalties” broadly are not new, and we clarified in Section 4.1 and the “What is novel in this work” paragraph that our contribution is layer-wise derivative control for compact surrogates, implemented via forward-mode Jacobian accumulation to regularize intermediate sensitivity amplification across depth—distinct from standard terminal $|\nabla_x f(x)|^2$ penalties and avoiding the numerical/memory burdens of second-order double-backprop in compact sweeps. We also broadened the literature discussion, including the PINNs connection suggested in your comments (Section 2), and updated the Introduction/Related Work with more recent sensitivity/smoothness control references.
>
> Table 1 ratios / last columns
> Following your suggestion, we now emphasize a scale-normalized tail-heaviness ratio (e.g., $\mathrm{p99}/\mathrm{mean}$) where appropriate, and we clarified the intended interpretation of tail metrics as “spikiness” indicators rather than only reporting single absolute tail values.
>
> We appreciate the feedback—these revisions significantly strengthen clarity, baseline context, and external validity of the empirical evaluation.

---

> > ### Comment · Reviewer_p54k · 2026-03-13
> >
> > Thank you for these replies.
> >
> > I have trouble finding them in the new version: please highlight changes in blue. Plus there are still sections that are small paragraphs (Section 2, Section 3, etc). Some are even shorter than before, while a thorough motivation and positioning is strongly needed.

---

> > > ### Author Response · Authors · 2026-03-16
> > > **Expanding Shorter Sections and Highlighting Alterations**
> > >
> > > Thank you for your feedback. We apologize for the difficulty in navigating the revisions. We have now uploaded a version where all substantial additions and modifications are highlighted in blue for clarity.
> > >
> > > Regarding the depth of motivation and positioning:
> > >
> > > Section 2 (Derivatives as Behavioral Control): We have significantly expanded this section to provide a theoretical and practical motivation for focusing on intermediate derivative amplification. It now includes a detailed discussion on the distinction between training stability and input sensitivity, as well as the conceptual link to Physics-Informed Neural Networks (PINNs) as suggested.
> > >
> > > Section 3 (Related Work): We have restructured this section to better position our work against recent literature (post-2020) and established baselines like Spectral Normalization and Sharpness-Aware Minimization (SAM).

---

### Decision · Action_Editor_R7fW · 2026-03-24

**Recommendation:** Reject

**Additional Comments:**

The recommendation is based on the reviewers' comments, the action editor's evaluation, and the authors’ response.

This submission should not be accepted in its current form due to several fundamental issues, as pointed out by the reviewers, including a lack of convincing experiments and evaluations (e.g., activation setups). While the rebuttal addresses some of the reviewers' concerns, the revised version still has several presentation and methodological issues to be worked out, which require significant revision and another round of full review. I hope the reviewers’ comments can help the authors prepare a better version of this submission.

**Audience:**

Yes

**Audience Explanation:**

yes.

**Claims And Evidence:**

No

**Claims Explanation:**

Not entirely. Some new experiments and evaluation need to be included to fully support the claims

**Resubmission Of Major Revision:**

The authors may consider submitting a major revision at a later time.